# Design and Feasibility Study of MRG–Based Variable Stiffness Soft Robot

**DOI:** 10.3390/mi13112036

**Published:** 2022-11-21

**Authors:** Luojing Huang, Hongsheng Hu, Qing Ouyang

**Affiliations:** 1College of Mechanical Engineering, Zhejiang University of Technology, Hangzhou 310023, China; 2College of Information Science and Engineering, Jiaxing University, Jiaxing 314001, China; 3School of Mechanical Engineering, Nanjing University of Science and Technology, Nanjing 210094, China; 4Taizhou Jiuju Technology Co., Ltd., Taizhou 225300, China

**Keywords:** MRG, halbach array, magnetic–air structure, variable stiffness, adaptivity

## Abstract

The conventional pneumatic soft robot has the problem of insufficient stiffness, while in the magnetorheological soft robot, the magnetic field provided by electromagnet has the disadvantage of oversized structure and poor flexibility. This paper presents a variable stiffness pneumatic soft robot based on magnetorheological grease (MRG) to solve these problems. Its three soft fingers cooperate with the adjustable gripper to adjust the gripping range for the robot hand, and it is used to provide gripping driving force through the bending drive. The MRG layer is designed on the gripping surface to provide adaptivity and rigid support for the gripped objects. A magnetic-air structure consisting of a Halbach array and Halbach array actuator is designed inside the soft fingers to provide a flexible magnetic field for the MRG layer. Theoretical and simulation analysis is carried out, and the results show that the state of the MRG changes and the stiffness of the clamping surface changes under the working pressure of 30 kPa. Finally, the experiment further proves the variable and high adaptivity of the surface stiffness of the gripping surface to reduce the damage to the gripped objects.

## 1. Introduction

Traditional robots are mainly rigid structures [1], and most components are made of rigid materials, such as metal, plastic, and other hard materials [2]. This makes it difficult to adapt to the complex external environment and has rigid contact with the grasped objects. In this case, it is difficult to control the gripping force and easily damages the surface of the objects under large loads [3]. Therefore, the traditional rigid structure robots are mostly designed to grasp fixed targets, rarely showing the same versatility and adaptivity as natural creatures [4]. With the continuous development of material science and robotics, researchers are gradually focusing on soft robotics. Compared to traditional rigid robots, soft robots have unlimited degrees of freedom, good flexibility and ductility, and excellent protection for working objects [5].

Soft robots are mainly driven by fluids or intelligent materials [6,7], including electroactive polymer actuators, shape memory alloy actuators [8], and other actuators. From the structural point of view, there are many kinds of soft robots, such as simple bag type [9], claw type [10,11], imitation human hand [12], and imitation creature [13]. The drive mode can be divided into pneumatic actuators [14], rope actuators [15], electric actuators [16], electrohydrodynamic pump actuators [17], magnetic actuators [18], thermal actuators, and mixed actuators. However, the soft robot has some advantages that rigid robots do not have but also has some shortcomings, such as poor surface stiffness and low stability.

In order to overcome the above limitations of soft robots, some scholars have conducted exploratory studies. G. Hwang et al. presented a reinforced soft gripper with a mechanically strengthened electroadhesion pad and a multi–layered dielectric elastomer actuator [16]. Mao et al. designed an eccentric actuator driven by stacked electrohydrodynamic pumps [19]. While providing a new flexural drive method, the stiffness characteristics of the actuators still need to be further enhanced. Researchers from Bishop–Moser Company [20] and Berlin University of Technology [21] have designed fiber–reinforced bending actuators. Fiber–reinforced bending actuators have a simpler tubular structure that simplifies the robotic design process. In addition, Pei et al. designed a soft actuator with a double–deformation cavity structure and a single cavity filled with particles [22]. Although these designs enhance the soft robot stiffness problem, there are still some shortcomings in terms of gripping surface adaptability and variable surface stiffness.

Magnetorheological smart materials include magnetorheological fluid (MRF), magnetorheological grease (MRG), and magnetorheological elastomer (MRE). MRF and MRG exhibit Newtonian flow behavior under ordinary environmental conditions and can accomplish millisecond state changes in a magnetic field environment. Under a certain magnetic field strength, the ferromagnetic particles in them can reach a magnetically saturated state and become more viscous [23]. Therefore, there is a great prospect for the flexible application of robots. In order to activate the magnetorheological material, a magnetic field needs to be generated. D. Hua et al. constructed the external magnetic field by means of an external Helmholtz coil [24]. However, this approach is more in the sense of exploration, and the practical engineering applicability is still lacking. J. Bernat et al. designed an electromagnetic coil–based magnetorheological elastomeric gripper [18]. A. Pettersson et al. used an electromagnetic coil to provide a magnetic field and designed a pocket gripper based on magnetorheological effects [25]. However, the electromagnetic coils were not suitable for delicate and flexible robotic applications due to structural and volume problems. Therefore, in this paper, the Halbach array, which has a simple structure, is chosen as the magnetic field generator based on the shortcomings of the above two types of magnetic field generators. The concept of Halbach permanent magnet array [26] was first proposed by Professor Klaus Halbach at Lawrence Berkeley National Laboratory, USA, as a new type of permanent magnet arrangement in which permanent magnets with different magnetization directions are arranged in a certain order so that the magnetic field on one side of the array is significantly enhanced while the other side is significantly weakened. Therefore, by changing the position of the Halbach array, the MRG can be provided with both a magnetic field environment and a zero–field environment, thus changing its stiffness state, simplifying the magnetic field setup, and increasing its flexibility.

Various ways to change the stiffness of flexible grippers, include fiber reinforcement, particle blocking, magnetorheological effects, and other methods. As shown in Table 1.

In this paper, a soft robot is designed, which contains three MRG–based variable stiffness pneumatic soft fingers, and innovatively designs a magnetic–air structure combining the H–A actuator (Halbach array actuator) and Halbach array. The mathematical model of the H–A actuator is constructed, and the motion of the outer cavity of the soft finger and the H–A actuator is simulated and analyzed. Finally, magnetic field simulation and experimental verification are used to study the stiffness characteristics under the magnetic field of the Halbach array, and the feasibility of variable stiffness of this design is analyzed.

## 2. Structural Design of The Soft Robot

In this paper, based on the characteristics of an ideal pneumatic actuator, combined with magnetorheological smart material technology, a soft robot is designed, consisting of two parts: a soft finger and an adjustable fixture, as shown in Figure 1. The soft finger is divided into a bending actuator, H–A actuator, Halbach array, and MRG layer. In terms of function, the bending actuator drives the soft robot hand to realize the bending action and gripping the objects; The H–A actuator drives the Halbach array to the working area where the MRG layer is located; The Halbach array provides a magnetic field for the MRG layer; The MRG layer is used as a contact surface to passively adapt to the shape of the item being gripped, providing enhanced gripping performance and adaptivity. In terms of material, the bending actuator and H–A actuator are made of 16° food–grade silicone (PS6600, Dongguan Chemical Material, Dongguan, China). The contact surface of the MRG layer is made of highly ductile latex, the air guide hole and holder are made of resin, and the permanent magnets in the Halbach array are N50 Nd–Fe–B magnets (Length, width, and height are 9.8 mm).

In the three–dimensional drawing of SolidWorks, the soft finger is attached to the slider, the slider is connected to the groove at the bottom of the rope reel through the traction rope, and the guide rail is designed with a compression spring, pulling the traction rope by turning the rope reel, compressing the spring, making the slider move and driving the three soft fingers to move inward.

In the whole working process of the fixture, rotating the rope reel pulling the rope to drive the three sliders to compress the spring and move to the center, used to adjust the range of clamping, and then fixed with the caliper, to achieve the purpose of adapting different sizes of objects. This is shown in Figure 2. Next, the bending actuator is driven by filling the air to bend the soft finger inward to pick up the object.

The fixture assists the soft robot in adjusting the clamping range, and soft fingers provide the active gripping force. The Halbach array and MRG layer work in tandem to provide variable stiffness to the clamped contact surface, providing strong stability, high adaptivity, and safety for objects of different shapes, sizes, and surface vulnerability.

## 3. Mathematical Model of Magnetic–Air Structure Theory

The core of this paper is the magnetic–air structure, and the key in this structure is the displacement of the Halbach array in the inner cavity of the soft finger. The H–A actuator provides the lateral displacement, and the bending actuator indirectly provides the longitudinal displacement while providing the clamping force, but the longitudinal displacement is smaller due to the space limitation of the inner cavity, so the influence is smaller. The lateral displacement is the main cause of the magnetic field distribution in the Halbach array. Therefore, in this paper, only the mechanical model of the H–A actuator is constructed to save time, and the effect of the longitudinal displacement produced by the bending actuator is verified through simulation and experiment.

The magnetic–air structure transports the Halbach array from the non–working area to the working area by the inflatable elongation of the H–A actuator. The material of the H–A actuator is silicone, as a class of superelastic materials, it has nonlinear mechanical characteristics, and it is difficult to use traditional constitutive equations to describe its stress–strain characteristics. Moreover, hyperelastic materials have unlimited degrees of freedom, high curvature, and high elongation. In order to show the nonlinear relationship between stress and strain, it needs to be described with a specific hyperelastic constitutive model. When creating a constitutive model of hyperelastic materials, two assumptions [27] are generally made: (1) isotropic; (2) incompressible. In the 1950s, Rivlin [28] proposed a strain energy density function for hyperelastic materials W as follows:(1)$$W=\overset{n}{\underset{\mathrm{ijk}=0}{\sum }}{\;C}_{\mathrm{ijk}}{\left(I_{1}-3\right)}^{i}{\left(I_{2}-3\right)}^{j}{\left(I_{3}-3\right)}^{k}$$

In Equation (1), $I_{1}$, $I_{2}$ and $I_{3}$ are the first, second, and third fundamental invariants of the Cauchy–Green [29] strain tensor, respectively, the expression is:(2)$$\{\begin{array}{c} I_{1}=\lambda_{1}^{2}+\lambda_{2}^{2}+\lambda_{3}^{2}\\ I_{1}=\lambda_{1}^{2}\lambda_{2}^{2}+\lambda_{2}^{2}\lambda_{3}^{2}+\lambda_{3}^{2}\lambda_{1}^{2}\\ I_{1}=\lambda_{1}^{2}\lambda_{2}^{2}\lambda_{3}^{2} \end{array}$$

In Equation (2), $\lambda_{1}$, $\lambda_{2}$, $\lambda_{3}$ are the main elongation ratios in the axial, radial and circumferential directions, respectively.
(3)$$\{\begin{array}{c} \lambda_{1}=\frac{L}{L_{0}}\\ \lambda_{2}=\frac{R_{1}^{\prime }}{R_{1}}\\ \lambda_{3}=\frac{L_{2}^{\prime }}{L_{2}} \end{array}$$

For hyperelastic materials, which are assumed to be incompressible, so that $I_{3}$ = 1, Equation (1) can be transformed into:(4)$$W=\overset{n}{\underset{\mathrm{ij}=0}{\sum }}{\;C}_{\mathrm{ij}}{\left(I_{1}-3\right)}^{i}{\left(I_{2}-3\right)}^{j}$$

Let n = 3, expand Equation (4), and simplify it accordingly to obtain the Yeoh model [30], which is applicable to simulate the large deformation behavior of hyperelastic materials and has a small fitting error between 40% and 150% of the strain [31], so the Yeoh model is used in this paper to establish the nonlinear relationship between material stress and strain. A typical density function binomial strain energy density function expression for the Yeoh model is as follows:(5)$$W=C_{10}\left(I_{1}-3\right)+C_{20}{\left(I_{1}-3\right)}^{2}$$

In Equation (5), $C_{10}$ and $C_{20}$ are material parameters, obtained by fitting the data from uniaxial tensile experiments [32].

The H–A actuator, which is the key of the structure, has a schematic sketch of its motion as shown in Figure 3a,b, and moves to the MRG layer after inflation.

In order to obtain the relationship between the length of each cell of the H–A actuator and the pressure of the charged air pressure, the cross–sectional deformation diagram of one of the chambers was analyzed. Theuninflated cross–section is shown in Figure 3c, and after inflation is shown in Figure 3d. The structural parameters are shown in Table 2.

By calculation, the internal volume of the air chamber and the wall volume of the air chamber are:(6)$$\{\begin{array}{c} V_{a}=\frac{2\pi }{3}L_{3}\left(R_{1}^{2}+R_{1}R_{2}+R_{2}^{2}\right)+\pi R_{1}^{2}L_{4}+2\pi R_{2}^{2}\left(L_{1}+L_{2}\right)\\ V_{b}=\pi R_{3}^{2}L_{5}+2\pi R_{4}^{2}-\pi R_{1}^{2}L_{4}-2\pi R_{2}^{2}\left(L_{1}+L_{2}\right) \end{array}$$

The process of axial displacement of the drive output is considered a quasi–static process, neglecting gravity and friction; according to the principle of imaginary work, the sum of the imaginary work done by all external forces in a static equilibrium system through the imaginary displacement is zero, the work done by air pressure should be balanced with the increased strain energy, and the specific expressions are as follows:(7)$$\sum W_{i}={\mathrm{PdV}}_{a}-V_{b}\mathrm{dW}=0$$

Combined with Equations (2), (3), and (5)–(7), and since the radial change of the actuator in this article is small, ignoring its effects, according to the incompressibility of the material, set $\lambda_{2}$ = 1, $\lambda_{3}$ = 1, you can obtain the function relationship between the driving air pressure P of the inner wall of the air cavity and the elongation amount L of the H–A actuator, expressed in F, the expression is as follows:(8)$$P=\frac{V_{b}\left(\frac{4C_{20}L^{3}}{L_{0}^{4}}-\frac{4C_{20}L}{L_{0}^{2}}+\frac{2C_{10}L}{L_{0}^{2}}\right)}{\frac{2\pi }{3}\left(R_{1}^{2}+R_{1}R_{2}+R_{2}^{2}\right)}=F\left(L\right)$$

The other parameters in function F other than P and L are constants, which are affected by $C_{10},{\;C}_{20}$, and parameters in Table 1.

## 4. Simulation Analysis of Soft Finger

The bending actuator mainly provides the drive and clamping force for the soft fingers, and the H–A actuator provides the drive for the Halbach array, and the simulation is used to simulate the motion of the actuator. Due to the large deformation and nonlinearity of the actuator, the ABAQUS is better at dealing with nonlinearity, so this paper uses it for the simulation of the bending actuator and H–A actuator.

### 4.1. Motion Simulation Analysis of Bending Actuators

The model of the bending actuator was created in 3D drawing software, and the actuator model was imported into ABAQUS. The entire model was built using tetrahedral secondary hybrid cells (C3D10H) with a cell count of 32189. The upper surface of the model is fixed, and an air pressure of 10 kPa is fed into the inner cavity to ensure a certain support strength so that the magnetic–air structure can reciprocate in it while providing a substrate for the MRG layer. Then an air pressure of 0~50 kPa was input to the bending actuator, the work was submitted for analysis, and the results are shown in Figure 4a. It can be analyzed from Figure 4a that after the air pressure of 10 kPa is input into the air chamber where the magnetic–air structure is located, the gripping contact surface of the soft finger has certain support, and then input different air pressure into the bending actuator. Due to the placement of the magnetic–air structure, the inner cavity is large, resulting in a small bending angle in the first half (0~30 kPa) and a gradual increase in the bending angle in the second half (30~50 kPa), the whole process makes the soft fingers bend in a certain range, providing gripping force, which can be used with adjustable clamps to achieve the goal of gripping items of different shapes and sizes.

### 4.2. Motion Simulation Analysis of H–A Actuator

Import the H–A actuator model into ABAQUS, and before importing it, the inflation holes and the connected Halbach array are removed to facilitate the calculation. Input the air pressure of 5~50 kPa into it, and obtain the deformation stress cloud diagram of the H–A actuator under different air pressures, as shown in Figure 4b. Combined with Figure 4b and theoretical calculation, the comparison curve shown in Figure 5 is obtained.

In Figure 5, it can be concluded from the comparison that the error value between the theory and model is smaller in the input air pressure range of 0~30 kPa, this is because the radial strain is ignored in the theoretical calculation. The input air pressure produces obvious radial strain after more than 30 kPa which leads to a larger error, so in the normal working range, the mathematical model of the H–A actuator can meet expectations and have reliability.

## 5. Characteristics of MRG under Halbach Array Magnetic Field

MRG is liquid in the general environment, so it can effectively change its contour according to gripped item’s shape when used as a gripping contact surface, thus better wrapping the object. When the magnetic field covers the MRG layer, the MRG will transform from a liquid to a solid–like state under the magnetorheological effect, thus increasing the support force and stiffness of the MRG layer.

### 5.1. Magnetic Field Analysis of Magnetic–Air Structures

#### 5.1.1. Halbach Array Magnetic Field Analysis

In this paper, a Halbach array is used to provide the magnetic field for the MRG layer, which is a new type of permanent magnet arrangement in which the permanent magnets are arranged in a certain order according to the magnetic flux direction so that the magnetic field on one side is significantly enhanced while the other side is significantly weakened. According to the permanent magnet arrangement proposed by Kim, W. J. [33], the arrangement of the Halbach array shown in Figure 6a is used in this paper.

The COMSOL is more accurate and convenient in the magnetic field simulation, so this paper uses it to analyze the magnetic field of the Halbach array. First, the magnetic fields generated by different permanent magnet arrays are simulated, and the results are compared, as shown in Figure 6c,d. The results of the magnetic field simulation show that the ordinary array has strict symmetry and the magnetic field strength is relatively strong at the two ends and weak at the sides and the middle, which cannot cover the MRG layer completely. In contrast, the magnetic field generated by the Halbach array has obvious unilateral enhancement and unilateral weakening effects, the magnetic field distribution is more uniform on the enhancement side, and the maximum magnetic field strength is increased by 1.8 times so that the enhancement side can cover the MRG layer and provide a magnetic field of sufficient strength.

#### 5.1.2. Magnetic Field Analysis of the Halbach Array and MRG Layer

According to the magnetic field strength and distribution of the Halbach array in the MRG layer, the corresponding magnetic field simulations were performed to simulate the magnetic field change in the MRG layer by moving the Halbach array due to the movement of the magnetic–air structure in the inner cavity, as shown in Figure 6b. Since the magnetic field of the Halbach array shows unilateral enhancement. However, the magnetic field distribution is not uniform; the magnetic field intensity is measured by taking nine points of a, b, c, d, e, f, g, h, and i, within the MRG layer in Figure 7 to roughly determine the magnetic field distribution within the MRG layer.

In order to make the magnetic field completely cover the MRG layer, the optimal position of the Halbach array needs to be determined. Therefore, the simulation analysis of the magnetic field strength at different distances is carried out for the lateral and longitudinal positions of the Halbach array and the MRG layer, respectively. The results of magnetic field simulations are shown in Figure 8.

In Figure 8a,c, with the continuous movement of the Halbach array, the farther the distance from the MRG layer is, the smaller the magnetic field strength of each measurement point and the smaller the slope of the curve. Therefore, we can conclude that there is no gap between the Halbach array and the MRG layer; that is, when $d_{1}$ = 0, the longitudinal direction is the optimal position.

During the movement, the magnetic field strengths of the measurement points at different positions of the MRG layer are also constantly changing. In Figure 8b,d, as the Halbach array keeps moving, the lateral distance between it and the MRG layer gets closer, and the magnetic field strength of each measurement point changes. Each measurement point is at a relatively high magnetic field strength when $d_{2}$ = 50. At this point, the Halbach array is in the optimal position in the lateral direction.

The fitted curves for nine of the measurement points are shown in Figure 8d; through data fitting, the approximate relationship between the moving distance of the Halbach array and the magnetic field strength in the MRG layer can be obtained as follows:(9)$$T=a_{0}+a_{1}\cos\left(\mathrm{dw}\right)+b_{1}\sin\left(\mathrm{dw}\right)+a_{2}\cos\left(2\mathrm{dw}\right)+b_{2}\sin\left(2\mathrm{dw}\right)+a_{3}\cos\left(3\mathrm{dw}\right)+b_{3}\sin\left(3\mathrm{dw}\right)$$
where the parameters of each curve are shown in Table 3.

When the Halbach array is in the optimal operating region, the magnetic field strength covering the MRG layer reaches its maximum value at this time. The magnetic field strength at each point needs to reach a certain value in order to excite the MRG, so the magnetic field strength of each measurement point within the MRG layer is simulated and analyzed, as shown in Figure 9. In the optimal state, the magnetic field inside the MRG bin shows a symmetrical distribution structure with a field strength range between 0.599 and 1.096, which can reach the magnetic saturation state of ferromagnetic particles in MRG; the minimum field strength can also cause the viscosity of the MRG to increase and reach a steady state. In this way, the MRG is excited to undergo a state change, which changes the stiffness of the soft robotic gripping surface.

### 5.2. Magnetic Field Distribution and MRG Layer Characteristics

#### 5.2.1. Magnetic Field Distribution of Different Permanent Magnet Arrangements

To verify the reliability of the magnetic field simulation results of the general array and the Halbach array, a set of comparison experiments is designed. The MRG was stirred well (the MRG used in this paper was MRG–70 made by Xinghui Wang of Nanjing University of Science and Technology; its main components are lubricating grease and ferromagnetic particles), poured into a transparent silicone recess. Different arrangements of permanent magnets were placed at the bottom center of the recess, and all other conditions were the same, and the magnetic field distribution conditions were observed in both cases, as shown in Figure 10.

Comparing Figure 10a and Figure 10b, in the general array permanent magnet environment, the magnetic fields on the top and bottom sides are symmetrical, and the number of ferromagnetic particles aggregated on both sides of the symmetry axis is similar., indicating that the magnetic field strengths on both sides are approximately equal; While in the Halbach array, the magnetic field lines on the upper side are more pronounced compared to Figure 10a and more ferromagnetic particles are gathered, while the magnetic field on the lower side is weak and the number of ferromagnetic particles is smaller. This set of comparative experiments proves that the Halbach array has the unilateral enhancement of magnetic field and the enhancement effect is significant, and also verifies the reliability of the simulation analysis.

#### 5.2.2. Variable Stiffness of the MRG Layer

Compare the magnetic field simulation design experiments to verify the characteristics of the MRG layer. The experimental bench shown in Figure 11 is built to compare the adaptability and stiffness changes of the MRG layer under different conditions. The MRG layer on top of the bench is a latex bag filled with MRG–70, and below is a bracket loaded with permanent magnet arrays. The bracket can be moved on an adjustable pallet, and moving the bracket can change the position of the permanent magnet arrays and control the moving distance by the number of teeth of the pallet, the layers on both sides of the bench, and the measuring tape, to precisely change the position of the covering magnetic field, simulate the real working conditions as much as possible and verify the stiffness characteristics of the MRG layer.

As shown in Figure 12a(i), the mass of the cube crystal is m:m=50 g, the length is $h_{1}$:$h_{1}=20\;\mathrm{mm}$, 1, 2, and 3 crystals are placed sequentially above the MRG layer, and with the increase of weight, the depression depth of the crystal under different magnetic field conditions is measured as h:$h=h_{1}-h_{2}$, to obtain the deformation resistance of MRG. In Figure 12a(ii–vi), the h value under different numbers of crystals is measured in the absence of magnetic field, general array magnetic field, and Halbach array magnetic field, and the results are shown in Figure 12b to verify the stiffness change of MRG under different magnetic field states. Depending on how the magnetic–air structure works, as the Halbach array moves through the inner cavity, the magnetic field covered by the MRG layer also changes, and the stiffness of the MRG layer changes as a result. In Figure 12a(iv–vi), the results are shown in Figure 12c by moving the bracket, comparing the depression depth H of the MRG layer in the lateral position, longitudinal position, and optimal position, and using this to verify the stiffness change of the MRG layer during the movement of the magnetic–air mechanism.

The experimental results in Figure 12b,c show that. Under the three magnetic field conditions with increasing magnetic field strength in Figure 12a(ii–vi), the depression depth of the MRG layer keeps decreasing, the viscosity of MRG increases, and the resistance to deformation of the MRG layer increases, thus proving that the stiffness of MRG layer keeps increasing. Moreover, K decreases as the weight increases under the Halbach array, which also indicates that the higher–weight MRG layer is more resistant to deformation. In Figure 12a(iv–vi), the depression depth of the MRG layer decreases sequentially at different positions of the Halbach array, which indicates that the resistance to deformation of the MRG layer is increasing and the stiffness of the Halbach array is increasing with the strengthening of the magnetic field strength covering the MRG layer in lateral and longitudinal movements.

#### 5.2.3. Adaptivity of the MRG Layer

In order to verify the adaptivity of the Halbach array at the optimal position for different items, three types of items with fragile surfaces, such as crystals, vegetables, and fruits, are selected for the MRG layer adaptivity experiments in this paper, as shown in Figure 13. When the Halbach array is in the optimal position, objects of different shapes and sizes are placed above the MRG layer under this magnetic field condition. The same force is applied above the objects, the objects are removed, and the MRG layer forms concave pits with the same outline as the objects and no longer changes, which can reflect the degree to which the MRG layer wraps the objects. Figure 13a presents a square profile at the bottom; Figure 13b presents an irregular–shaped profile; Figure 13c presents a triangular profile; Figure 13d presents an elliptical profile; Figure 13e presents two adjacent circular profiles; Figure 13f presents a columnar profile. The comparison shows that the MRG layer has good adaptivity to different objects under this condition, and the stiffness of the MRG layer has been enhanced at this time, and according to the contour formed by the MRG layer, the irregular side shape of the object can be adaptively wrapped when the robot clamps the object.

### 5.3. Soft Gripper Gripping Surface Variable Stiffness Feasibility Analysis

Based on the theoretical calculations and the simulation analysis in the magnetic–air structure, the relationship between the input air pressure and the magnetic field strength in this structure can be derived by combining Equations (8) and (9), as shown in Figure 14. After the input of air pressure to the soft robot, the magnetic field strength of the measurement points near the Halbach array grows rapidly as the H–A actuator drives the Halbach array to the MRG layer. As the input air pressure increases to 30 kPa, the Halbach array remains in the same vertical bisector as the MRG layer, and the magnetic field completely covers the MRG layer; the magnetic field strength at each measurement point is kept within a certain range, stabilized, and the magnetic field strength is 0.58~1.072T, which can reach the magnetic saturation state of ferromagnetic particles in MRG and make the viscosity of MRG increase.

Combined with the experimental results, after the soft robot input air pressure, during the movement of the magnetic–air structure, as the Halbach array moves laterally and longitudinally, the magnetic field covering the MRG layer gradually increases, and the stiffness of the MRG layer becomes larger and larger. When the soft robot grips the object, the bending actuator provides the clamping force, which is gradually embedded in the MRG layer under the influence of the clamping force and the increasing magnetic field strength. When the air pressure gradually increases to 30 kPa, the magnetic–air structure reaches the optimal position. The stiffness of the MRG layer reaches the maximum, which can provide an upward support force for the bottom of the object embedded in it (as shown in the schematic diagram in the lower right corner of Figure 14) to achieve variable stiffness in the process of clamping the object and provide it with upward support force without damaging the surface of the object. It is demonstrated that the soft robot designed in this paper is feasible in terms of variable stiffness of clamping surface, which can provide theoretical, simulation, and experimental basis for further design of the soft robot.

## 6. Discussion and Conclusions

In this paper, we propose an MRG–based soft robot with an adjustable fixture and bending actuator for grasping; the magnetic–air structure designed internally has the unique advantages of simple structure, small size, and easy control compared with the electromagnetic coil, and through the combination with MRG, the surface stiffness is changed. The adaptivity is improved while maintaining support, which provides a new direction for solving the stiffness problem of a soft robot. In order to enable the magnetic field generated by the Halbach array to cover the MRG layer, a theoretical model of the H–A actuator was established, and the actuator was simulated and compared with the theoretical results. Magnetic field simulation and experiments verify that changes in the magnetic field affect the state of the MRG, thereby changing the stiffness of the clamping surface and improving its adaptivity. Since the theoretical model ignores the small radial strain for simplicity in the calculation and ignores the friction and other factors in the simulation analysis to save time, there may be a slight discrepancy in the simulation results. The experiments only simulate part of the actual working conditions, and the theoretical model and simulation need to be further optimized.

It is demonstrated by theoretical model and simulation analysis that the magnetic–air structure can move the magnetic field generated by the Halbach array to the MRG layer working area when the input air pressure is 30 kPa. Based on the results of the magnetic field simulation, it was determined that the magnetic field on the enhanced side of the Halbach array increased by a factor of 1.8, and the magnetic field strength ranged from 0.58 to 1.072 T, which achieves the minimum magnetic field intensity for activating MRG–70. Through comparative experiments, it is verified that the change of magnetic field can change the stiffness of the MRG layer and improve the resistance to deformation. Under the unilateral enhanced magnetic field when the Halbach array is in the optimal position, the MRG layer has high adaptability to different shapes and sizes of surface fragile items, while the increased stiffness can provide upward support for the bottom of the gripper, which has a broad application background in flexible gripping. Next, we will further optimize the theoretical model and simulation, process and fabricate the prototype of the soft robot, complete the grasping experiment under real working conditions, further verify the reliability of the design, improve the practical application value, and provide new ideas for the future development of the soft robot.

## Figures and Tables

**Figure 1 micromachines-13-02036-f001:**
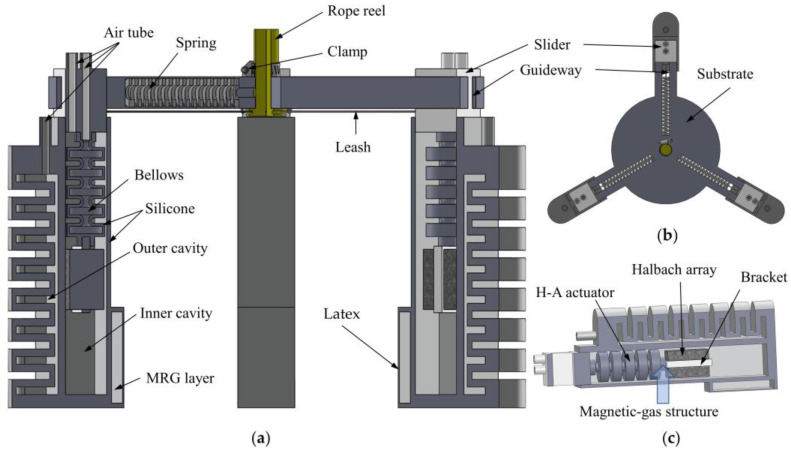
Three–dimensional rendering of the soft gripper. (**a**) Structure diagram of the overall soft robot; (**b**) Top view of soft finger fixture holder; (**c**) Internal cutaway view of the soft finger.

**Figure 2 micromachines-13-02036-f002:**
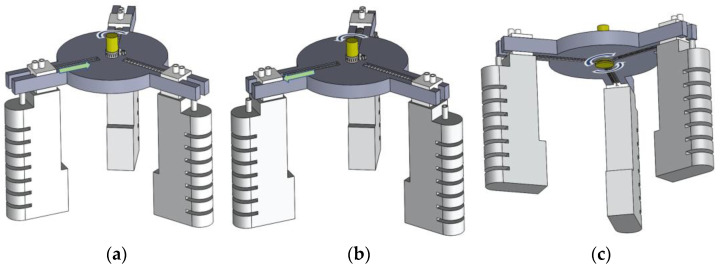
Assembly drawing of the fixture. (**a**) Fixture contraction movement. (**b**) Fixture expansion movement. (**c**) Rope reel rotating rope movement. The rotation arrow represents the direction of rotation of the rope reel, and the straight arrow represents the direction of movement of the part.

**Figure 3 micromachines-13-02036-f003:**
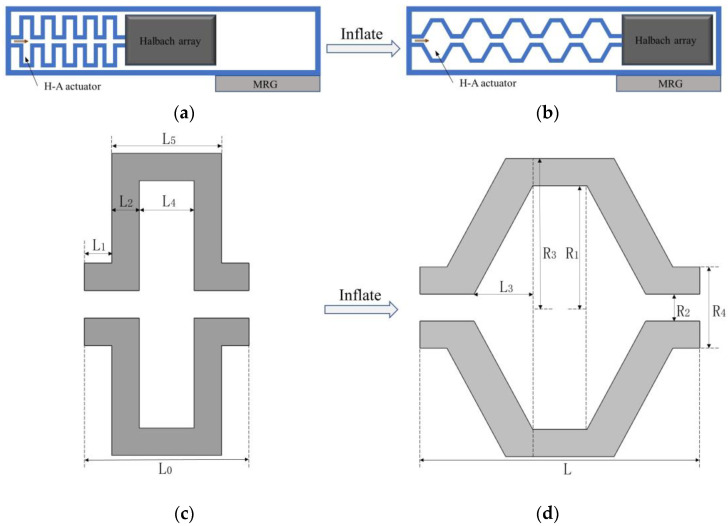
Schematic of the H–A actuator motion process and simplified diagram of an air chamber in an H–A actuator. (**a**,**c**) The state before inflation; (**b**,**d**) The state after inflation.

**Figure 4 micromachines-13-02036-f004:**
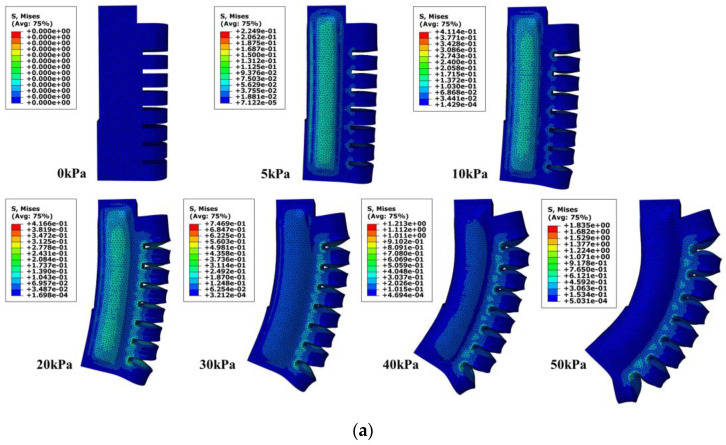
Stress cloud diagram of the actuator under various air pressure conditions in the range of 0~50 kPa. (**a**) Bending actuator (**b**) H–A actuator.

**Figure 5 micromachines-13-02036-f005:**
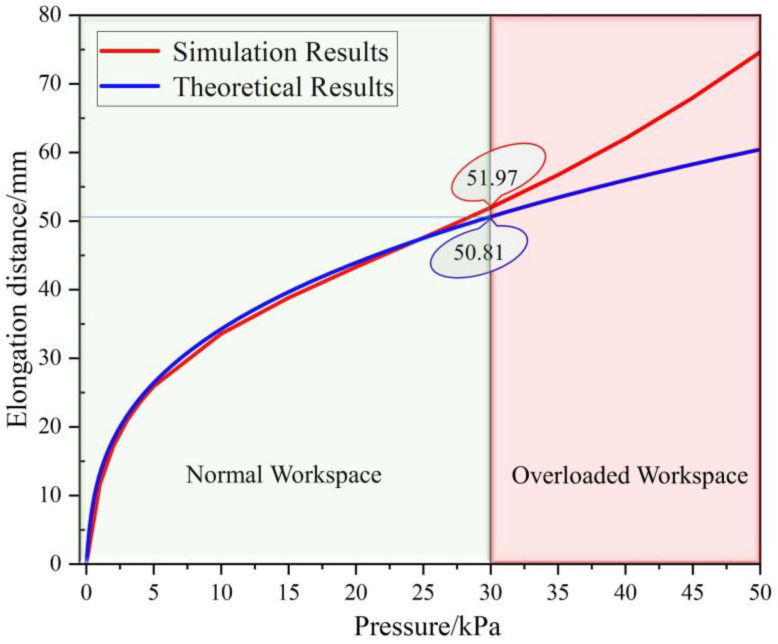
Comparison between the results of the simulation and the results of the numerical calculation.

**Figure 6 micromachines-13-02036-f006:**
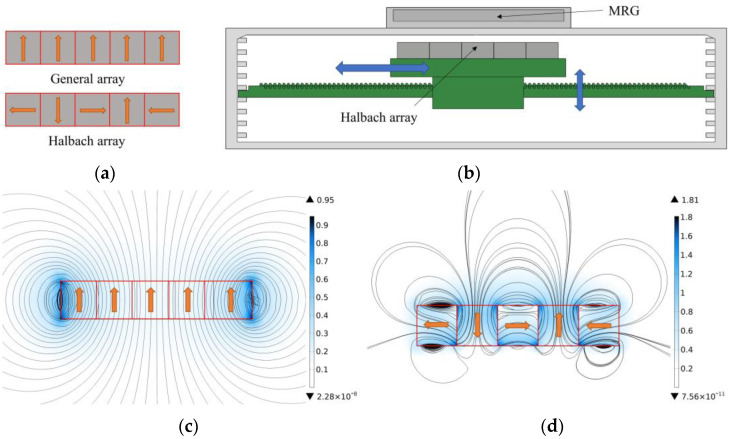
The effect of the arrangement and distance of the permanent magnets on the MRG layer. (**a**) Two different arrays of permanent magnets (the arrow indicates the direction of movement); (**b**) The effect of the lateral and longitudinal movement of the Halbach array on the MRG layer; (**c**) Magnetic field effect diagram of an ordinary array of permanent magnets; (**d**) Magnetic field rendering of Halbach array. Where the direction of the arrow is the direction of the magnetic flux of the permanent magnet.

**Figure 7 micromachines-13-02036-f007:**
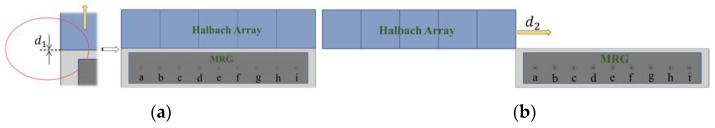
Schematic of Halbach array and MRG layer. (**a**) Longitudinal movement schematic; (**b**) Lateral movement schematic. Where the yellow arrow is the direction in which the Halbach array moves, $d_{1}$ is the longitudinal moving distance between the two and $d_{2}$ is the lateral moving distance between the two.

**Figure 8 micromachines-13-02036-f008:**
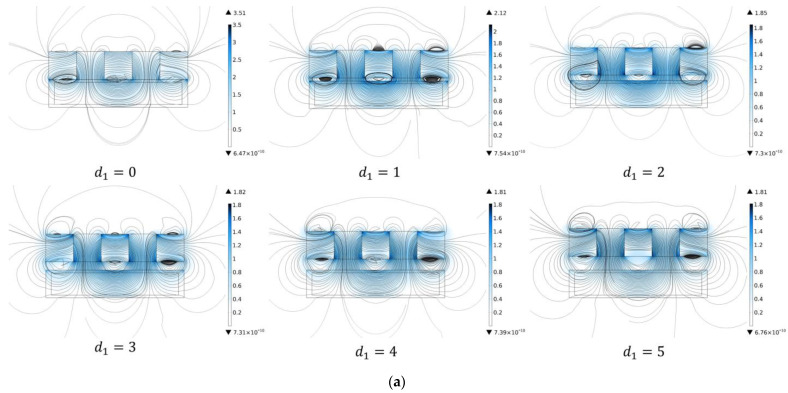
(**a**) Magnetic field distribution at $d_{1\;}$= 0; 1; 2; 3; 4; 5. (**b**) Magnetic field distribution at $d_{2}$ = 0; 10; 20; 30; 40; 50. (**c**) Magnetic field intensity curves for different longitudinal distances of Halbach array and MRG layers. (**d**) Magnetic field strength curves for Halbach array and MRG layer lateral distance.

**Figure 9 micromachines-13-02036-f009:**
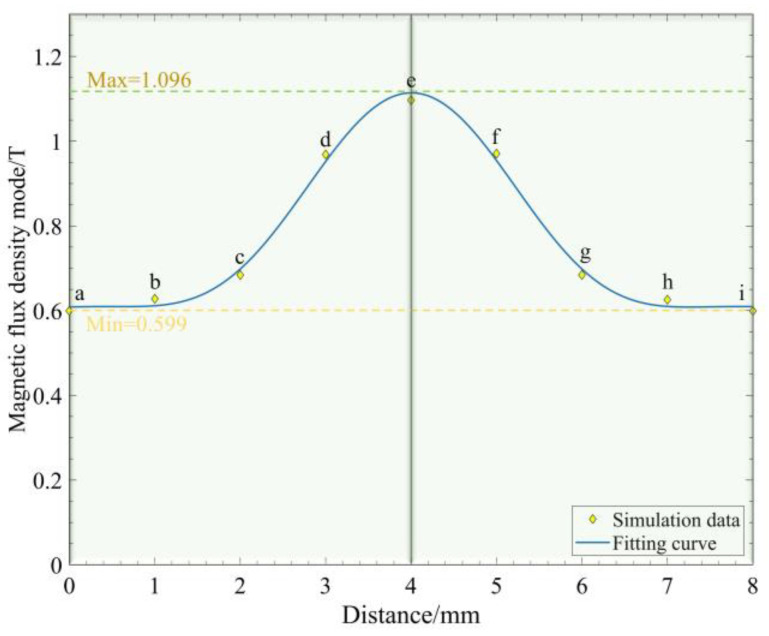
Magnetic field strength curves for Halbach array and MRG layer optimal working area. Where a, b, c, d, e, f, g, h, i are the magnetic field strength measurement points in the MRG layer.

**Figure 10 micromachines-13-02036-f010:**
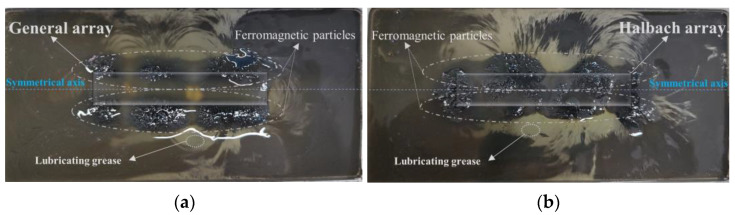
Comparison of the magnetic fields of different arrangements of permanent magnets. (**a**) General array. (**b**) Halbach array.

**Figure 11 micromachines-13-02036-f011:**
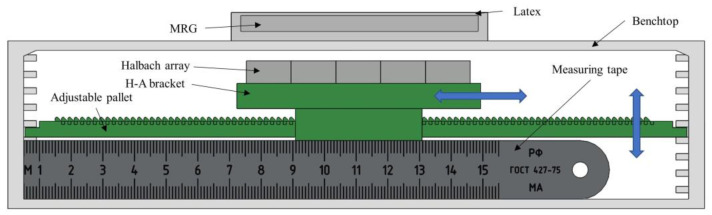
Experimental bench.

**Figure 12 micromachines-13-02036-f012:**
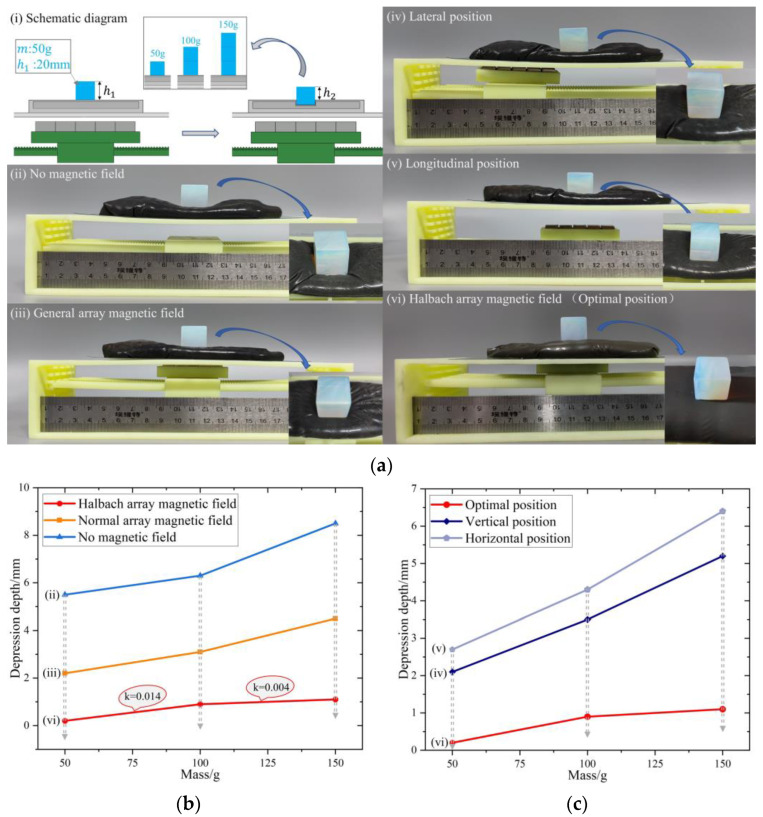
(**a**) Variable stiffness experiment. (**b**) The depth of depression of the MRG layer under different magnetic field conditions (k is the slope). (**c**) The depth of depression of the MRG layer at different locations. (**i**) Schematic diagram, (**ii**) No magnetic field, (**iii**) General array magnetic field, (**iv**) Lateral position, (**v**) Longitudinal position, (**vi**) Halbach array magnetic field (Optimal position).

**Figure 13 micromachines-13-02036-f013:**
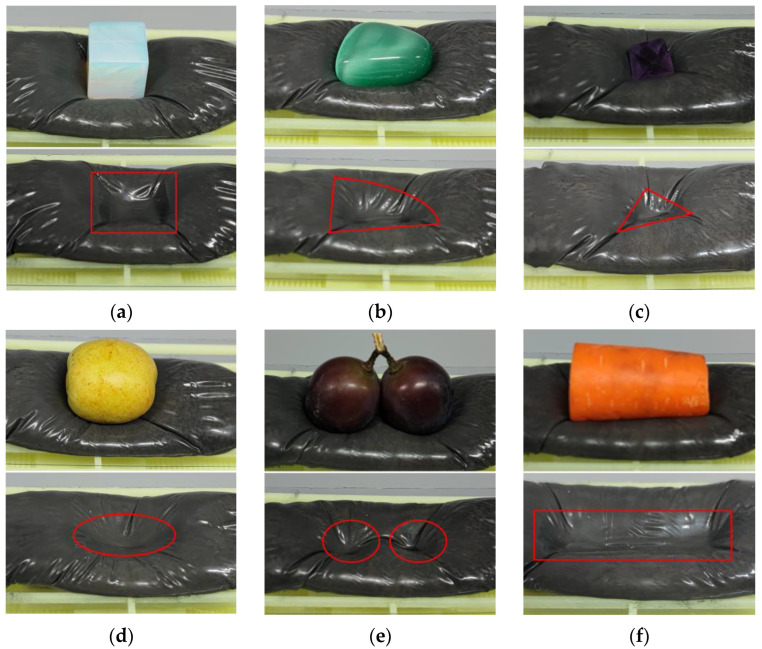
Adaptivity of different objects. (**a**) Cube Crystal. (**b**) Irregular Crystal. (**c**) Polyhedral Crystal. (**d**) Jujube. (**e**) Grape. (**f**) Carrot. The concave pits are marked with red lines.

**Figure 14 micromachines-13-02036-f014:**
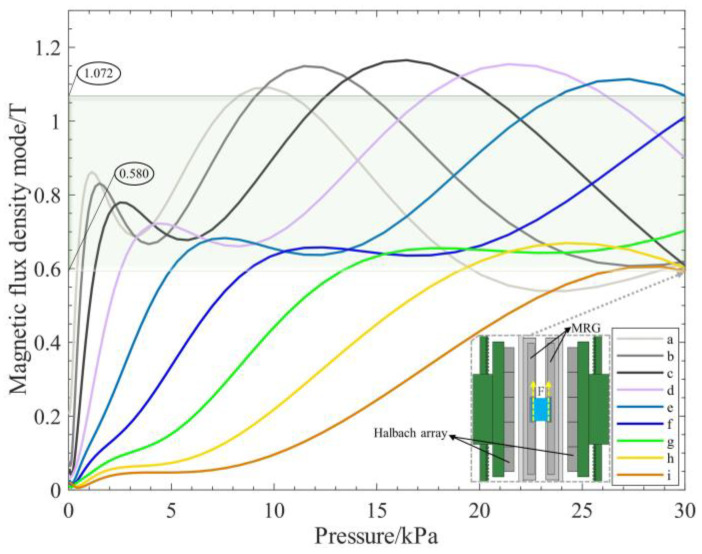
The soft robot input air pressure and the magnetic field strength curve of the MRG layer. Where a, b, c, d, e, f, g, h, i are the measurement points. Inside the dotted line is a schematic diagram of the gripped object, and F is the supporting force at the bottom of the gripped object.

**Table 1 micromachines-13-02036-t001:** Comparison of variable stiffness methods.

			Magnetorheological Effects		
Characteristics	Fiber Reinforcement	Particle Blocking	Electromagnetic Coli–MRF	Electromagnetic Coil–MRE	Halbach Array–MRG
Overall volume	Middle size	Large size	Large size	Middle size	Small size
Response speed	Fast	Average	Fast	Average	Fast
Stiffness range	Small	Large	Small	Small	Average
Adaptivity	Low	Average	High	Low	High
Difficulty of realization	Simple	Simple	Difficult	Difficult	Simple

**Table 2 micromachines-13-02036-t002:** Structural parameters of the air chamber.

Symbol	Parameter	Value/mm
$L_{1}$	Small inner cavity half–length	1.5
$L_{2}$	Sidewall thickness	1.2
$L_{4}$	Large inner cavity internal width	4.6
$L_{5}$	Large inner cavity external width	7
$R_{1}$	Large internal cavity inner diameter	10.3
$R_{2}$	Small internal cavity inner diameter	3.3
$R_{3}$	Large internal cavity volume increase	11.5
$R_{4}$	Small inner cavity outer diameter	4.5

**Table 3 micromachines-13-02036-t003:** Parameter values of each fitted curve.

	a	b	c	d	e	f	g	h	i
$a_{0}$	0.674	0.665	0.593	0.538	0.491	0.915	0.458	0.007	0.246
$a_{1}$	0.262	0.332	0.413	0.371	0.245	0.346	0.236	0.273	0.368
$b_{1}$	0.058	0.032	0.171	0.277	0.379	1.044	0.412	0.277	1.978
$a_{2}$	0.026	0.058	0.182	0.187	0.141	0.558	0.111	0.333	0.147
$b_{3}$	0.004	0.012	0.048	0.015	0.104	0.013	0.062	0.053	0.158
$a_{3}$	0.176	0.163	0.078	0.061	0.071	0.019	0.081	0.050	0.244
$b_{3}$	0.067	0.134	0.041	0.075	0.086	0.228	0.074	0.113	0.395
w	0.095	0.092	0.101	0.098	0.097	0.070	0.072	0.068	0.002

## Data Availability

The original contributions presented in the study are included in the article, and further inquiries can be directed to the corresponding author.

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
