# Peer review of "Design and Feasibility Study of MRG–Based Variable Stiffness Soft Robot"

_micromachines, 2022, doi:10.3390/mi13112036_

Round 1

Reviewer 1 Report

The authors designed the MRG-based soft robot with variable stiffness with a magnetic-air structure inside. They built up a theoretical model and simulation analysis that the magnetic-air structure can move the magnetic field generated by the Halbach array to the MRG layer working area when the input air pressure is 30kPa. This job extends the pneumatic and magnetic methods in soft robot fields. 

1. in the introduction, the authors overlook the hydraulic methods using functional fluids. like a Fluidic rolling robot using voltage-driven oscillating liquid and an Eccentric actuator driven by stacked electrohydrodynamic pumps. they should consider those. 

2. In their demonstration, their design is like a jumping gripper. However, their structure can not grip the object in the vertical direction. Therefore, what is the usage of magnetorheological grease? 

3. Figure.1 shows a three-dimensional rendering of the soft gripper and figure.2 shows an assembly drawing of the fixture. I suggest the authors show the reader the optical images of this setup.

4. the authors seem to have a variable stiffness. From their paper, how did they derive this conclusion?

5. Figure.22, what is the meaning of the legend?

Author Response

Thank you for your comments.

Reviewer 2 Report

Authors have carried out a number of simulation and experimental work towards the design and Feasibility Study of MRG-based Variable Stiffness Soft Robot. The paper could be accepted for publication in this journal after the following minor revisions.

1.       The flowchart in Figure 16 is not quite explanatory and the flow could be improved, or the flowchart removed to avoid confusing the reader.

2.       Instead of having too many figures, some figures could be merged as one. For instance, figure 9 and Figure 10, also Figure 19 and 20 etc. Check other instances of this and merge the figure

3.       A table comparing this work and the literature would be helpful to understand more about the research gap filled.

4.       What would be the effect of grasping a metallic object or when working with objects having embedded magnets?

5.       Figure 17 should be labelled to show the magnetic field being compared. i.e.  Add texts to Figure 17 to improve it’s understanding.

6.       Section 6 which with the title “Discussion and Conclusion”, contains only the conclusion and no discussion per se.

Author Response

Thank you for your comments.
